# Predictors of severe or lethal COVID-19, including Angiotensin Converting Enzyme inhibitors and Angiotensin II Receptor Blockers, in a sample of infected Italian citizens

**Francesca Bravi**[1☯], **Maria Elena Flacco**[2☯], **Tiziano Carradori**[1], **Carlo Alberto Volta**[1,3], **Giuseppe Cosenza**[4], **Aldo De Togni**[4], **Cecilia Acuti Martellucci**[5], **Giustino Parruti**[6], **Lorenzo Mantovani**[7,8], **Lamberto Manzoli**[1,2]*

1 "Sant'Anna" University Hospital of Ferrara, Ferrara, Italy, 2 Department of Medical Sciences, University of Ferrara, Ferrara, Italy, 3 Department of Morphology, Surgery and Experimental Medicine, University of Ferrara, Ferrara, Italy, 4 Local Health Authority of Ferrara, Ferrara, Italy, 5 Department of Biomedical Sciences and Public Health, University of the Marche Region, Ancona, Italy, 6 Local Health Authority of Pescara, Cerveteri, Italy, 7 Center for Public Health Research, University of Milan—Bicocca, Milan, Italy, 8 IRCCS Multimedica, Sesto San Giovanni, Italy

☯ These authors contributed equally to this work.
* lmanzoli@post.harvard.edu

**Data Availability Statement:** All relevant data are within the paper and its Supporting Information files.

## Abstract

### Aims

This retrospective case-control study was aimed at identifying potential independent predictors of severe/lethal COVID-19, including the treatment with Angiotensin-Converting Enzyme inhibitors (ACEi) and/or Angiotensin II Receptor Blockers (ARBs).

### Methods and results

All adults with SARS-CoV-2 infection in two Italian provinces were followed for a median of 24 days. ARBs and/or ACEi treatments, and hypertension, diabetes, cancer, COPD, renal and major cardiovascular diseases (CVD) were extracted from clinical charts and electronic health records, up to two years before infection. The sample consisted of 1603 subjects (mean age 58.0y; 47.3% males): 454 (28.3%) had severe symptoms, 192 (12.0%) very severe or lethal disease (154 deaths; mean age 79.3 years; 70.8% hypertensive, 42.2% with CVD). The youngest deceased person aged 44 years. Among hypertensive subjects (n = 543), the proportion of those treated with ARBs or ACEi were 88.4%, 78.7% and 80.6% among patients with mild, severe and very severe/lethal disease, respectively. At multivariate analysis, no association was observed between therapy and disease severity (Adjusted OR for very severe/lethal COVID-19: 0.87; 95% CI: 0.50–1.49). Significant predictors of severe disease were older age (with AORs largely increasing after 70 years of age), male gender (AOR: 1.76; 1.40–2.23), diabetes (AOR: 1.52; 1.05–2.18), CVD (AOR: 1.88; 1.32–

**Funding:** The authors received no specific funding for this work

**Competing interests:** The authors have declared that no competing interests exist.

2.70) and COPD (AOR: 1.88; 1.11–3.20). Only gender, age and diabetes also predicted very severe/lethal disease.

## Conclusion

No association was found between COVID-19 severity and treatment with ARBs and/or ACEi, supporting the recommendation to continue medication for all patients unless otherwise advised by their physicians.

## Introduction

Novel coronavirus disease (COVID-19) is spreading worldwide, and has caused over 250,000 deaths so far [1]. The mortality rate varies widely by age and across individuals, ranging from 0.2% among healthy, young-adults, to >10% among older persons with pre-existing conditions [1].

Although the pharmacological treatment was not assessed, the first observational studies on patients with severe disease reported a high prevalence of comorbidities that are often treated with angiotensin converting enzyme (ACE) inhibitors, such as cerebrovascular diseases, coronary heart disease, hypertension and diabetes [2–4]. Observing that human pathogenic coronaviruses bind their target cells through angiotensin-converting enzyme 2 (ACE2) [5–8], and that a few studies reported an increase in ACE2 expression mediated by angiotensin II type-I receptor blockers (ARBs) and ACE inhibitors (more consistently on animals than in humans) [9–16], some hypothesized that the increased expression of ACE2 would facilitate infection with Severe Acute Respiratory Syndrome Coronavirus 2 (SARS-CoV-2), thus the hypertension treatment with ACE2-stimulating drugs, as well as ACE2 polymorphisms, might increase the risk of developing severe COVID-19 [17–19]. Consequently, this would lead to a serious conflict regarding treatment, because ACE2 reduces inflammation and has been suggested as a potential new therapy for inflammatory lung diseases, cancer, diabetes, and hypertension [17, 20–23]. In the wake of two preliminary cohort studies reporting a lower [24] or similar [25] COVID-19 mortality among inpatients hypertensive subjects treated with ARBs and ACE inhibitors, the potential predictors of COVID-19 and of disease severity, including anti-hypertensive medications, were recently analyzed by a few observational studies [26–28]. With one exception [27], no increased risk emerged from the use of ARBs or ACE inhibitors; however, the role of other potentially linked predictors, including age and cardiovascular comorbidities [17, 29, 30], differed across the population analyzed, and still requires confirmation. We have performed a case-control study on all SARS-CoV-2 infected subjects diagnosed in two Italian provinces, retrieving admission and pharmacological data up to two years before infection, in order to confirm the potential independent predictors of severe/lethal COVID-19, including treatment with ACE inhibitors and/or ARBs.

## Materials and methods

This case-control, retrospective study compared the proportion of subjects treated with ARBs and/or ACE inhibitors among three groups of subjects with SARS-CoV-2 infection:

a. asymptomatic infection or mild disease, defined as fever or malaise plus at least one of the followings: sore throat, muscle pain, shortness of breath, dry cough, headache, conjunctivitis, and diarrhea [31], with no hospital admission;

b.  severe disease, requiring hospital admission, not in an intensive care unit;

c.  very severe or lethal disease, requiring admission in an intensive care unit and/or causing death.

The sample includes all subjects with diagnosis of infection made in the Province of Ferrara, up to April 2, and the Province of Pescara, Italy, up to April 24, 2020, by the Central Laboratory of the University Hospital of Ferrara or the Central Laboratory of the Pescara Hospital (and confirmed by the National Institute of Health). All diagnoses were made using (real time) reverse transcription polymerase chain reaction (rRT-PCR) on oropharingeal specimens. The assays were those originally proposed by the Charité-Universitätsmedizin Berlin Institute of Virology [32], and then endorsed by the WHO [33].

The data on background pharmacological treatment up to the previous two years (January 1, 2018) were obtained from the National database of drug prescription, and integrated with clinical chart information for hospitalized subjects. Data have been collected on the following drugs: ACE inhibitors (ATC classes: C09A and C09B), ARBs (C09C and C09D), and insulin or other anti-diabetic drugs (A10). Information on age, gender, and pre-existing conditions of all subjects were obtained through data-linkage with hospital discharge abstracts (Italian SDO), which have been queried from the day of the diagnosis until January 1st, 2015. All admission data have been revised manually by two physicians (LM and MEF) and the following conditions have been included in the analyses: malignant tumors, major cardiovascular diseases (heart failure, myocardial infarction and stroke—CVD), type II diabetes, renal disease and chronic obstructive pulmonary diseases (COPD, bronchitis, pneumonia, asthma, and emphysema).

The study complies with the Declaration of Helsinki, the research protocol was approved by the Ethics Committee of the Emilia-Romagna Region (code 287, approved on March 24, 2020), and the requirement for informed consent was waived because of the retrospective and pseudo-anonymized nature of the data.

First, the differences across groups with mild, severe or very severe/lethal disease were evaluated using two-way ANOVA with post-hoc Tukey HSD test for continuous variables, and Mantel-Haenzsel chi-squared test for categorical ones. A sample restricted to hypertensive subjects was used to compare the subjects treated and untreated with ACE inhibitors or ARBs.

Multivariate logistic regression was used to investigate the potential independent predictors of severe or very severe/lethal COVID-19. Four models were built, two were restricted to hypertensive subjects (A and B), and two included the total sample (Models C and D) and. Models A and C were fit to predict severe or very severe disease (grouped together), while Models B and D to predict very severe/lethal disease only (and repeated to predict death, with similar results, which have not been shown to avoid redundancy). All recorded variables (age, gender, CVD, diabetes, renal disease, cancer and COPD) were included a priori in all models, with the exception of treatments with ACE inhibitors and ARBs, that were excluded from Models C and D because of multicollinearity with hypertension. Standard diagnostic procedures were adopted to check all models validity: influential observation analysis (Dbeta, change in Pearson chi-square and similar), multicollinearity, interaction terms, Hosmer-Lemeshow test for the goodness of fit and C statistic (area under the Receiving Operator Curve). Statistical significance was defined as a two-sided p-value<0.05, and all analyses were carried out using Stata, version 13.1 (Stata Corp., College Station, TX, 2014).

## Power analysis

Our sample of 129 hypertensive subjects with severe/lethal COVID-19, and 414 hypertensive subjects with mild disease or asymptomatic infection, had 80% statistical power to detect a difference of 20% or higher (corresponding to a relative risk ≥1.20) in the risk of severe/lethal disease between users of ARBS or ACE inhibitors (exposed group), and non users (controls).

## Results

The sample consisted of 1603 subjects (mean age 58.0y; 47.3% males): 454 (28.3%) had severe symptoms, 192 (12.0%) very severe or lethal disease (Table 1). The sample of infected subjects, compared with the available estimates of the general population of the two provinces [34], showed higher prevalence of diabetes (12.1% in the sample versus ≅5% in the general population); hypertension (33.9% versus ≅17%), COPD (6.0% versus ≅3%) and CVD (16.1% versus ≅3%). 154 subjects deceased (mean age 79.3 years; 54.6% males); of them, 70.8% were hypertensive, 42.2% were diagnosed a CVD; 27.9% diabetics. Twenty subjects with very severe disease were younger than 60 years; the youngest being a male aged 33 years. Of those deceased, eight were younger than 60 years, and the youngest was a female aged 44 years.

At univariate analysis, as compared to the subjects with mild disease, those with severe or very severe/lethal disease were significantly more likely to be older, diabetics, hypertensive, diagnosed with COPD, CVD, and renal disease, and treated with ARBs and/or ACE inhibitors (all p<0.05). Among hypertensive subjects (n = 543), however, the proportion of those treated with ARBs or ACE inhibitors (whose medication was not discontinued during the follow-up)

**Table 1. Characteristics of the sample.**

| | Overall sample | Mild | Severe | Very severe/lethal | p [A] | p [B] | p [C] |
|---|---|---|---|---|---|---|---|
| **Variables** | **(n = 1603)** | **(n = 957)** | **(n = 454)** | **(n = 192)** | | | |
| Mean age in years (SD) | 58.0 (20.9) | 50.4 (20.2) | 66.4 (16.9) | 76.2 (12.9) | <0.001 | <0.001 | <0.001 |
| Male gender, % | 47.3 | 42.5 | 53.1 | 57.3 | <0.001 | <0.001 | 0.3 |
| Diabetes, % | 12.1 | 6.8 | 16.5 | 28.1 | <0.001 | <0.001 | 0.001 |
| COPD, % | 6.0 | 2.9 | 9.2 | 14.1 | <0.001 | <0.001 | 0.070 |
| Cancer, % | 7.6 | 5.1 | 10.1 | 14.1 | <0.001 | <0.001 | 0.2 |
| Major cardiovascular diseases [D], % | 16.1 | 6.9 | 26.9 | 36.5 | <0.001 | <0.001 | 0.015 |
| Renal disease, % | 5.4 | 2.4 | 8.8 | 12.0 | <0.001 | <0.001 | 0.2 |
| Hypertension, % | 33.9 | 21.6 | 45.6 | 67.2 | <0.001 | <0.001 | <0.001 |
| Antihypertensive treatment with: | | | | | | | |
| • ACE inhibitors, % | 15.7 | 11.2 | 19.4 | 29.2 | <0.001 | <0.001 | 0.006 |
| • ARBs, % | 14.2 | 9.0 | 19.8 | 27.1 | <0.001 | <0.001 | 0.042 |
| • ACE inhibitors or ARBs, % | 28.1 | 19.1 | 35.9 | 54.2 | <0.001 | <0.001 | <0.001 |
| **Sample restricted to hypertensive subjects** | (n = 543) | (n = 207) | (n = 207) | (n = 129) | | | |
| Antihypertensive treatment with: | | | | | | | |
| • ACE inhibitors, % | 46.2 | 51.7 | 42.5 | 43.4 | 0.061 | 0.14 | 0.9 |
| • ARBs, % | 42.0 | 41.6 | 43.5 | 40.3 | 0.7 | 0.8 | 0.6 |
| • ACE inhibitors or ARBs, % | 82.9 | 88.4 | 78.7 | 80.6 | 0.008 | 0.049 | 0.7 |

[A] Mild versus severe subjects.

[B] Mild versus very severe subjects

[C] Severe versus very severe subjects.

[D] Congestive heart failure, myocardial infarction, or stroke. COPD = Chronic obstructive pulmonary diseases. ACE = Angiotensin-Converting Enzyme.

ARBs = Angiotensin II Receptor Blockers.

**Table 2.** **Logistic regression model predicting severe or very severe/lethal COVID-19 syndrome (grouped together, Model A) or very severe/lethal disease only (Model B), in the sample restricted to hypertensive subjects.**

| | (A) Severe or very severe/lethal | p* | (B) Very severe/lethal | p* |
|---|---|---|---|---|
| Antihypertensive drugs: | OR (95% CI) | | OR (95% CI) | |
| •ACE Inhibitors | 0.70 (0.44–1.13) | 0.15 | 0.82 (0.49–1.36) | 0.4 |
| •ARBs | 0.91 (0.56–1.47) | 0.7 | 0.83 (0.50–1.40) | 0.5 |
| •ACE Inhibitors or ARBs * | 0.58 (0.34–1.01) | 0.056 | 0.87 (0.50–1.49) | 0.6 |

ACE = Angiotensin-Converting Enzyme. ARBs = Angiotensin II Receptor Blockers. OR = odds ratio. CI = Confidence interval. All estimates have been adjusted for age, gender, diabetes, major cardiovascular diseases, COPD, cancer and renal diseases.

* Models A and B have been repeated including ACE or ARBs treatment, grouped, with no substantial changes for other independent variables.

were higher among those with asymptomatic/mild disease (88.4%, versus 78.7% and 80.6% among patients with severe and very severe/lethal disease, respectively—Table 1).

In multivariable analyses restricted to hypertensive subjects (Models A and B, Table 2), the treatment with ARBs and/or ACE inhibitors never increased the likelihood of severe or very severe/lethal disease (all p>0.405).

The significant predictors of severe disease were male gender (Adjusted Odds Ratio—AOR: 1.76; 95% Confidence Interval—CI: 1.40–2.23), diabetes (AOR: 1.52; 1.05–2.18), CVD (AOR: 1.88; 1.32–2.70) and COPD (1.88; 1.11–3.20), and older age, which showed an exponential increase after 70 years: compared with the subjects younger than 50 years, the AORs of those aged 70–79 and ≥80 years were 5.72 (3.81–8.58) and 9.06 (6.04–13.6), respectively (Model C; Table 3). Only male gender, older age and diabetes also predicted very severe/lethal disease (Model D; Table 3).

**Table 3.** **Logistic regression model predicting severe or very severe/lethal COVID-19 syndrome (grouped together, Model C) or very severe/lethal disease only (Model D), in the total sample (n = 1603).**

| | (C) Severe or very severe/lethal | p* | (D) Very severe/lethal | p* |
|---|---|---|---|---|
| Variables | OR (95% CI) | | OR (95% CI) | |
| Male gender | 1.76 (1.40–2.23) | <0.001 | 1.69 (1.20–2.37) | 0.003 |
| Age class, in years | | | | |
| •<50 (Ref. cat.) | 1 | – | 1 | – |
| •50–59.9 | 2.36 (1.68–3.32) | <0.001 | 3.92 (1.48–10.3) | 0.006 |
| •60–69.9 | 3.51 (2.43–5.07) | <0.001 | 11.4 (4.63–28.1) | <0.001 |
| •70–79.9 | 5.72 (3.81–8.58) | <0.001 | 16.5 (6.66–40.9) | <0.001 |
| •≥80 | 9.06 (6.04–13.6) | <0.001 | 27.1 (11.1–66.3) | <0.001 |
| Diabetes | 1.52 (1.05–2.18) | 0.025 | 1.58 (1.06–2.34) | 0.023 |
| Hypertension | 1.10 (0.82–1.47) | 0.5 | 1.39 (0.94–2.05) | 0.097 |
| Major cardiovascular diseases ** | 1.88 (1.32–2.70) | 0.001 | 1.05 (0.71–1.56) | 0.8 |
| Cancer | 1.26 (0.81–1.95) | 0.3 | 1.11 (0.67–1.82) | 0.7 |
| COPD | 1.88 (1.11–3.20) | 0.020 | 1.44 (0.84–2.47) | 0.2 |
| Renal disease | 1.58 (0.90–2.76) | 0.11 | 1.13 (0.64–1.99) | 0.7 |

COPD = Chronic obstructive pulmonary diseases. OR = odds ratio. CI = Confidence interval. ACE = Angiotensin-Converting Enzyme. ARBs = Angiotensin II Receptor Blockers.

* Treatments with ACE inhibitors and ARBs have not been included in both models because of multicollinearity with hypertension.

** Heart failure, myocardial infarction, or stroke.

## Discussion

The main findings of this retrospective, observational study, are the following: first, it is confirmed that, among hypertensive subjects, the use of ACE inhibitors or ARBs up to two years preceding SARS-CoV-2 infection did not affect the severity of COVID-19. Second, older age, male gender, diabetes, and the presence of COPD or CVD were independent predictors of severe disease, with a sharp increase of risk among subjects older than 70 years. Third, only older age, male gender and diabetes were associated with a higher likelihood of very severe/lethal disease.

Several hypotheses have been made on the association between COVID-19 progression and treatment with ACE inhibitors and ARBs [20–22, 35, 36]. On one side, some asked whether the therapy should be discontinued during SARS-Cov-2 pandemic [17, 37] because COVID-19 was strongly associated with hypertension, which is frequently treated with ARBs and ACE inhibitors [2–4]. It was indeed hypothesized that: (a) ACE2 up-regulation mediated by ARBs (and, to a lesser extent, by ACE inhibitors) might increase patients' susceptibility to SARS--CoV-2 entry into host cells and further viral propagation [18, 19], (b) virus binding to ACE2 might reduce its activity, thus leading to increased levels of Angiotensin II and consequent pulmonary vasoconstriction, inflammation and oxidative organ damage, and increased risk of acute lung injury [20]. On the opposite side, other scientists suggested that, other than being harmful, ARBs and ACE inhibitors use in patients with cardiovascular risk factors and known or suspected COVID-19 may even exert a beneficial effect, as ACE2 up-regulation could increase the conversion of Angiotensin II to Angiotensin-(1–7), a peptide with potentially protective anti-inflammatory properties [35, 36].

Recently, a few large observational studies based upon in- and outpatient electronic health records [26–28] examined the association between antihypertensive medications and the risk of COVID-19 and/or a severe/lethal disease: our results are in line with most of the previous findings on an absence of risk with ACE inhibitors and/or ARBs use. In brief, the studies by Mancia et al [26], and Reynolds et al [28] reported no difference in COVID-19 severity or death between treated and untreated subjects, for both ARBs and ACE inhibitors. Instead, the results by Mehta et al [27] were partially discordant: the study found no association between treatment with ARBs and death, but those treated with ACE inhibitors showed a significantly higher risk of ICU admission. In our sample, in order to further investigate the potential beneficial effects of ACE inhibitors, the impact of which might be larger in patients with diabetes, COPD, or cardiovascular diseases, we performed additional analyses, stratified by comorbidities. We found however no significant differences in the risk of severe/lethal COVID-19 among treated and untreated patients with either CVD, or diabetes and COPD (data not shown).

Besides sample size and provenance, there were few differences between this and previous studies: all studies included all infected subjects, hospitalized or not, evaluated disease progression beyond mortality, and retrieved medications and admissions from electronic health records (with the exception of Mehta et al, who assessed the medications exclusively at the time of testing for SARS-CoV-2 [27]).

Overall, the present and previous findings confirm those from preliminary Chinese cohorts [24, 31], and although confirmation from randomized studies is required [38], they strongly support the statements of several experts [39, 40] and scientific societies, including the European Medicines Agency [41], the European Society of Cardiology [42], and the American Heart Association [43], who recommend continuation of ARBs or ACE inhibitors medication for all patients, unless otherwise advised by their physicians.

With regard to the role of the other risk factors that have been suggested for severe COVID-19, including age, male gender, hypertension, diabetes, COPD, and major cardiovascular diseases, it has been correctly argued that, so far, available data were unadjusted, thus the relative importance of underlying health conditions was unclear [21, 29]. In this study, we found support for a potential role of gender, diabetes, COPD and CVD, beyond age, in COVID-19 progression to a severe disease, whereas only gender and diabetes significantly increased the risk of a lethal or very severe outcome. Thus, the present study confirms prior findings on the independent relationship of older age and male gender with death [44], and of COPD with progression towards severe disease [26]. Instead, at least two issues may have influenced the conflicting results on the role of CVD and COPD in predicting very severe/lethal disease (an association showed in some prior populations [44]—but lacking in the present as well as in other recent findings [26]): first, the relatively small sample of the present study; second, a marked difference in the population here enrolled, as compared to previous studies which included randomly selected SARS-CoV2 negative subjects as controls [26]. Given the present scenario, further population-based cohort studies, with longer follow-up are clearly needed [29] to clarify these findings.

In addition to a relatively small sample, a limitation of the present study is the lack of tobacco smoking and body mass index among the variables that have been recorded, because we could not extract such information for half of the deceased subjects, as well as for many of those that were not hospitalized. Other limitations are the lack of an evaluation of the severity of the underlying cardiovascular diseases, and the absence of data on other antihypertensive medications. However, their potential role in altering the relationship between ARBS or ACE inhibitors and risk of severe/lethal COVID-19 remains unclear: none of the previous studies on the topic assessed cardiovascular diseases severity [26–28], and the two studies that included the use of other antihypertensive drugs into multivariable analyses did not find substantial differences between the adjusted and unadjusted relative risks of death [26, 28].

In conclusions, the present study did not find any association between COVID-19 severity and treatment with ARBs, ACE inhibitors, or both, and confirms previous findings in supporting the recommendation of several scientific societies to continue ARBs or ACE inhibitors medication for all patients, unless otherwise advised by their physicians, who should thus be reassured. The adjusted analyses substantially confirm prior reports, indicating that the risk of severe or lethal COVID-19 largely and significantly increases among the elderly, males, diabetics, and those with COPD or major cardiovascular diseases.

## Supporting information

**S1 Appendix. Database.**
(XLS)

## Author Contributions

**Conceptualization:** Francesca Bravi, Maria Elena Flacco, Tiziano Carradori, Lamberto Manzoli.

**Data curation:** Francesca Bravi, Maria Elena Flacco, Lamberto Manzoli.

**Formal analysis:** Francesca Bravi, Maria Elena Flacco, Cecilia Acuti Martellucci, Lamberto Manzoli.

**Investigation:** Carlo Alberto Volta, Giuseppe Cosenza, Aldo De Togni, Cecilia Acuti Martellucci, Giustino Parruti.

**Methodology:** Maria Elena Flacco, Carlo Alberto Volta, Giuseppe Cosenza, Aldo De Togni, Cecilia Acuti Martellucci, Giustino Parruti, Lorenzo Mantovani, Lamberto Manzoli.

**Project administration:** Francesca Bravi, Tiziano Carradori.

**Supervision:** Tiziano Carradori, Lamberto Manzoli.

**Validation:** Tiziano Carradori, Carlo Alberto Volta, Lorenzo Mantovani, Lamberto Manzoli.

**Writing – original draft:** Maria Elena Flacco, Lamberto Manzoli.

**Writing – review & editing:** Francesca Bravi, Tiziano Carradori, Carlo Alberto Volta, Giuseppe Cosenza, Aldo De Togni, Giustino Parruti, Lorenzo Mantovani.

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
