## [Decision Letter · Decision Letter 0]

26 May 2020

PONE-D-20-13065

Predictors of severe or lethal COVID-19, including Angiotensin Converting Enzyme Inhibitors and Angiotensin II Receptor Blockers, in a sample of infected Italian citizens.

PLOS ONE

Dear Dr. Manzoli,

Thank you for submitting your manuscript to PLOS ONE. After careful consideration, we feel that it has merit but does not fully meet PLOS ONE’s publication criteria as it currently stands. Therefore, we invite you to submit a revised version of the manuscript that addresses the points raised during the review process.

Besides novelty, which is not an imoportant scope of our journal, there are some issues to be revised as pointed out by two experts.

We look forward to receiving your revised manuscript.

Kind regards,

Tatsuo Shimosawa, M.D., Ph.D.

Academic Editor

PLOS ONE

Journal Requirements:

2. In your Methods section, please provide additional information about the participants included in the analysis . Please ensure you have provided sufficient details to replicate the analyses such as: a)  a description of any inclusion/exclusion criteria that were applied to participant inclusion, b) a statement as to whether your sample can be considered representative of a larger population.

Reviewers' comments:

Reviewer's Responses to Questions

**Comments to the Author**

1. Is the manuscript technically sound, and do the data support the conclusions?

Reviewer #1: Yes

Reviewer #2: Yes

2. Has the statistical analysis been performed appropriately and rigorously? 

Reviewer #1: No

Reviewer #2: I Don't Know

3. Have the authors made all data underlying the findings in their manuscript fully available?

Reviewer #1: Yes

Reviewer #2: Yes

4. Is the manuscript presented in an intelligible fashion and written in standard English?

Reviewer #1: Yes

Reviewer #2: Yes

5. Review Comments to the Author

Reviewer #1: 1. Please clarify the novelty of this study. Especially, how about discussing a little bit more the strength and shortness of this study compared with a report by Mehra et al.?

2. If the data are available, how about comparing the prevalence of hypertension, CVD, and diabetes between the diseased subjects and the whole residence of Provinces of Ferrara and Pescara or Italy?

3. Table 1. Please use 2-way ANOVA followed by post-hoc analysis for the comparison among 3 groups stratified by the COVID-19 severity.

4. Please clarify each variable included in multivariate logistic regression analysis, instead of the current description: “All recorded variables were included a priori in all models, …”

5. Figure 1 may not be necessary.

6. Lines 170-171. The gist of the description is unclear and misleading.

7. Ref 5,6 were not adequate.

Reviewer #2: The authors conducted retrospective case-control study in Italian citizens trying to identify potential independent predictors of severe/lethal COVID-19, and showed no association between COVID-19 severity and treatment with ARBs and/or ACE inhibitors, supporting the previous recommendation to continue the medication. This is an important topic, but it was unfortunate not to see any new findings.

The diagnosis of COVID-19 should be confirmed by PCR tests. Although they described where the laboratory diagnosis was done, they did not mention PCR tests at all. They should provide detailed information of PCR tests.

The worse symptoms of cardiovascular diseases would increase the opportunity to use ARBs or ACE inhibitors. Normalization of the data with disease severity would be necessary. In addition, treatment with ACE inhibitors or ARBs should be compared with other anti-hypertensive drugs.

The authors compared with or without ARBs / ACE inhibitors in hypertension. Sometimes diseases are overlapped. They should clarify whether diabetes or COPD are overlapped with cardiovascular diseases. RAS inhibitors on diabetes patients might have protective impacts on COVID-19 severity.

It is interesting to see the ACE Inhibitors or ARBs treatment showing a trend for reduced odds ratio (Table 2). Beneficial or adverse effects of the drugs would have to be examined more rigorously. For instance, was prescription or medication continued during the infection for each patient?

For references, they cited recent short review or commentary on COVID-19 but mostly ignored key original papers. This is not acceptable. There are nearly two decades of research on ACE2 and SARS-CoV accumulated, and both basic and clinical papers of origin should be cited. The reference should be extensively revised.

6. PLOS authors have the option to publish the peer review history of their article (what does this mean?). If published, this will include your full peer review and any attached files.

Reviewer #1: No

Reviewer #2: No

---

## [Author Response · Author response to Decision Letter 0]

29 May 2020

Answers to Referee I comments

I-1. The Referee wrote: “Please clarify the novelty of this study. Especially, how about discussing a little bit more the strength and shortness of this study compared with a report by Mehra et al.?"

We agree that, unfortunately, the results are not novel. Please acknowledge, however, that the evidence on SARS-CoV-2 is emerging so rapidly that a finding becomes old after weeks. Please believe that we are the first to be sorry for that. We also entirely agree that the strengths and weaknesses of the study should have been discussed further, comparing it with Mehra et al. and the other observational studies on the same topic (which now also include Mehta et al., JAMA Cardiol 2020). Accordingly, to comply also with the suggestions of the first reviewer, we entirely revised the Discussion section. In specific, the previous paragraphs:

"Recently, three large observational studies based upon inpatient only [14] or in- and outpatient electronic health records [15, 16] examined the association between antihypertensive medications and the risk of COVID-19 and/or a severe/lethal disease: our results substantially confirm these previous findings and, concordantly, no increased risk emerged with ACE inhibitors and/or ARBs use. Of note, Mehra et al [14], after controlling for age, gender, current smoke and comorbidities, found similar death rates among COVID-19 patients using ARBs, and even lower death rates among ACE inhibitors users. The latter results, however - although in line with a previous observational analysis on Chinese in-hospital COVID-19 subjects [12], are based upon hospitalized patients only, may be due to unmeasured confounding and, in the absence of randomized evidence, should be considered only preliminary [23].

Our study, likewise Mancia et al [15] and Reynolds at al [16], included all infected subjects (either hospitalized or not), retrieved all medications and admissions from electronic health records, and evaluated the disease progression beyond mortality. Concordantly, all these findings confirm those from preliminary Chinese cohorts [12, 19] and strongly support the statements of several experts [24, 25] and scientific societies, including the European Medicines Agency [26], the European Society of Cardiology [27], and the American Heart Association [28], who recommend continuation of ARBs or ACE inhibitors medication for all patients, unless otherwise advised by their physicians."

have been replaced by the followings:

"Recently, a few large observational studies based upon inpatient only [26] or in- and outpatient electronic health records [27-29] examined the association between antihypertensive medications and the risk of COVID-19 and/or a severe/lethal disease: our results are in line with most of the previous findings on an absence of risk with ACE inhibitors and/or ARBs use. In brief, the studies by Mancia et al [27], and Reynolds et al [28] reported no difference in COVID-19 severity or death between treated and untreated subjects, for both ARBs and Ace inhibitors. Instead, the results by Mehra et al [26] and Mehta et al [29] were partially discordant: both studies found no association between treatment with ARBs and death, but those treated with ACE inhibitors showed a significantly higher risk of ICU admission in the study by Mehta et al [29], and a significantly lower risk of in-hospital death in the study by Mehra et al. [26]. In our sample, in order to further investigate the potential beneficial effects of ACE inhibitors, the impact of which might be larger in patients with diabetes, COPD, or cardiovascular diseases, we performed additional analyses, stratified by comorbidities. We found however no significant differences in the risk of severe/lethal COVID-19 among treated and untreated patients with either CVD, or diabetes and COPD (data not shown).

Besides sample size and provenance, the main differences between this and previous studies pertain the inclusion criteria (we included all subjects, hospitalized or not, such as Mancia et al [27], Reynolds et al [28], Mehta et al [29], and unlike Mehra et al [26]); the retrieval of all medications and admissions from electronic health records (unlike Mehta et al [29]), and the evaluation of disease progression beyond mortality (unlike Mehra et al [26]).

Overall, the present and previous findings confirm those from preliminary Chinese cohorts [24, 32], and although confirmation from randomized studies is required [39], they strongly support the statements of several experts [40, 41] and scientific societies, including the European Medicines Agency [42], the European Society of Cardiology [43], and the American Heart Association [44], who recommend continuation of ARBs or ACE inhibitors medication for all patients, unless otherwise advised by their physicians."

I-2. The Referee wrote: “If the data are available, how about comparing the prevalence of hypertension, CVD, and diabetes between the diseased subjects and the whole residence of Provinces of Ferrara and Pescara or Italy?"

We agree and accordingly added the following sentences in the Results section: "The sample of infected subjects, compared with the available estimates of the general population of the two provinces [34], showed higher prevalence of diabetes (12.8% in the sample versus �5% in the general population); hypertension (33.9% versus �17%), COPD (6.0% versus �3%) and CVD (16.1% versus �3%). Please acknowledge that, although the estimates of the two provinces were quite similar (with the exception of cardiovascular diseases), the overall estimate from the provinces was derived computing a weighted mean of the latest available data (2018) from the Italian Institute of Statistics. 

I-3. The Referee wrote: “Table 1. Please use 2-way ANOVA followed by post-hoc analysis for the comparison among 3 groups stratified by the COVID-19 severity".

We agree and we accordingly used two-way ANOVA followed by Tukey honestly significant difference test. Methods and Table footnotes were updated, while the Results did not differ. 

I-4. The Referee wrote: “Please clarify each variable included in multivariate logistic regression analysis, instead of the current description: “All recorded variables were included a priori in all models, …”

We agree and we accordingly added that age, gender, CVD, diabetes, renal disease, cancer and COPD were included a priori in all models.

I-5. The Referee wrote: “Figure 1 may not be necessary".

We agree and accordingly cut Figure 1.

I-6. The Referee wrote: “Lines 170-171. The gist of the description is unclear and misleading."

We agree and we accordingly revised the text. The following sentences:

"On one side, some asked whether the therapy should be discontinued during SARS-Cov-2 pandemic [7, 22] because of a strong association between hypertension and disease severity (although the pharmacological treatment were not assessed) [2-4], and based on the hypothesis that (a) ACE2 up-regulation mediated by ARBs (and, to a lesser extent, by ACE inhibitors) might increase patients' susceptibility to SARS-CoV-2 entry into host cells and further viral propagation [20], (b) virus binding to ACE2 might reduce its activity, thus leading to increased levels of Angiotensin II and consequent pulmonary vasoconstriction, inflammation and oxidative organ damage, and increased risk of acute lung injury [8]"

were replaced with

"On one side, some asked whether the therapy should be discontinued during SARS-Cov-2 pandemic [17, 38] because COVID-19 was strongly associated with hypertension, which is frequently treated with ARBs or ACE inhibitors [2-4]. It was indeed hypothesized that (a) ACE2 up-regulation mediated by ARBs (and, to a lesser extent, by ACE inhibitors) might increase patients' susceptibility to SARS-CoV-2 entry into host cells and further viral propagation [18, 19], (b) virus binding to ACE2 might reduce its activity, thus leading to increased levels of Angiotensin II and consequent pulmonary vasoconstriction, inflammation and oxidative organ damage, and increased risk of acute lung injury [20]."

I-7. The Referee wrote: “Ref 5,6 were not adequate."

We agree, and we accordingly replaced references n. 5-6 with the following (now numbered as [5-8]):

5. Hoffmann M, Kleine-Weber H, Schroeder S, et al. SARSCoV-2 cell entry depends on ACE2 and TMPRSS2 and is blocked by a clinically proven protease inhibitor. Cell 2020;181:271-80.

6. Kuba K, Imai Y, Rao S, Gao H, Guo F, Guan B, et al. A crucial role of angiotensin converting enzyme 2 (ACE2) in SARS coronavirus-induced lung injury. Nat Med. 2005;11(8):875-9.

7. Li W, Moore MJ, Vasilieva N, et al. Angiotensin-converting enzyme 2 is a functional receptor for the SARS coronavirus. Nature 2003;426: 450-4.

8. Xiao X, Chakraborti S, Dimitrov AS, Gramatikoff K, Dimitrov DS. The SARS-CoV S glycoprotein: expression and functional characterization. Biochem Biophys Res Commun. 2003;312(4):1159-64

Please also acknowledge that the sentences in the Introduction previously reported as: 

"Observing that human pathogenic coronaviruses bind their target cells through angiotensin-converting enzyme 2 (ACE2) [5], and that the expression of ACE2 is substantially increased in diabetics and patients who are treated with ACE inhibitors and angiotensin II type-I receptor blockers (ARBs) [5,6], Fang and colleagues hypothesized that the increased expression of ACE2 would facilitate infection with Severe Acute Respiratory Syndrome Coronavirus 2 (SARS-CoV-2), thus the hypertension treatment with ACE2-stimulating drugs, as well as ACE2 polymorphisms, might increase the risk of developing severe COVID-19 [7]"

were replaced as follows:

"Observing that human pathogenic coronaviruses bind their target cells through angiotensin-converting enzyme 2 (ACE2) [5-8], and that a few studies reported an increase in ACE2 expression mediated by angiotensin II type-I receptor blockers (ARBs) and ACE inhibitors (more consistently on animals than in humans) [9-16], some hypothesized that the increased expression of ACE2 would facilitate infection with Severe Acute Respiratory Syndrome Coronavirus 2 (SARS-CoV-2), thus the hypertension treatment with ACE2-stimulating drugs, as well as ACE2 polymorphisms, might increase the risk of developing severe COVID-19 [17-19]."

We accordingly added the following new references:

9. Furuhashi M, Moniwa N, Mita T, Fuseya T, Ishimura S, Ohno K, et al. Urinary angiotensin-converting enzyme 2 in hypertensive patients may be increased by olmesartan, an angiotensin II receptor blocker. Am J Hypertens. 2015;28(1):15-21.

10. Luque M, Martin P, Martell N, Fernandez C, Brosnihan KB, Ferrario CM. Effects of captopril related to increased levels of prostacyclin and angiotensin-(1-7) in essential hypertension. J Hypertens. 1996;14(6):799-805.

11. Zisman LS, Keller RS, Weaver B, Lin Q, Speth R, Bristow MR, et al. Increased angiotensin-(1-7)-forming activity in failing human heart ventricles: evidence for upregulation of the angiotensin-converting enzyme Homologue ACE2. Circulation. 2003;108(14):1707-12.

12. Ferrario CM, Jessup J, Chappell MC, Averill DB, Brosnihan KB, Tallant EA, et al. Effect of angiotensin-converting enzyme inhibition and angiotensin II receptor blockers on cardiac angiotensin-converting enzyme 2. Circulation. 2005;111(20):2605-10.

13. Gallagher PE, Ferrario CM, Tallant EA. MAP kinase/phosphatase pathway mediates the regulation of ACE2 by angiotensin peptides. Am J Physiol Cell Physiol. 2008;295(5):C1169-74.

14. Igase M, Strawn WB, Gallagher PE, Geary RL, Ferrario CM. Angiotensin II AT1 receptors regulate ACE2 and angiotensin-(1-7) expression in the aorta of spontaneously hypertensive rats. Am J Physiol Heart Circ Physiol. 2005;289(3):H1013-9.

15. Ishiyama Y, Gallagher PE, Averill DB, Tallant EA, Brosnihan KB, Ferrario CM. Upregulation of angiotensin-converting enzyme 2 after myocardial infarction by blockade of angiotensin II receptors. Hypertension. 2004;43(5):970-6.

16. Soler MJ, Ye M, Wysocki J, William J, Lloveras J, Batlle D. Localization of ACE2 in the renal vasculature: amplification by angiotensin II type 1 receptor blockade using telmisartan. Am J Physiol Renal Physiol. 2009;296(2):F398-405.

18. Esler M, Esler D. Can angiotensin receptor-blocking drugs perhaps be harmful in the COVID-19 pandemic? J Hypertens 2020;38:781-2.

19. Sommerstein R, Grani C. Preventing a COVID-19 pandemic: ACE inhibitors as a potential risk factor for fatal COVID-19. BMJ 2020;368: m810.

Answers to Referee II comments

II-1. The Referee wrote: “The authors conducted retrospective case-control study in Italian citizens trying to identify potential independent predictors of severe/lethal COVID-19, and showed no association between COVID-19 severity and treatment with ARBs and/or ACE inhibitors, supporting the previous recommendation to continue the medication. This is an important topic, but it was unfortunate not to see any new findings. The diagnosis of COVID-19 should be confirmed by PCR tests. Although they described where the laboratory diagnosis was done, they did not mention PCR tests at all. They should provide detailed information of PCR tests."

We entirely agree that his information should have been reported, and we are sorry for the oversight. We accordingly added the following sentences in the Methods section: "All diagnoses were made using (real time) reverse transcription polymerase chain reaction (rRT-PCR) on oropharingeal specimens. The assays were those originally proposed by the Charité-Universitätsmedizin Berlin Institute of Virology [33], and then endorsed by the WHO [34]".

The following references were thus added:

33. Corman VM, Landt O, Kaiser M, Molenkamp R, Meijer A, et al. (2020) Detection of 2019 novel coronavirus (2019-nCoV) by real-time RT-PCR. Euro Surveill 25.

34. World Health Organization (2020) Laboratory testing for coronavirus disease (COVID-19) in suspected human cases - Interim guidance.

II-2. The Referee wrote: “The worse symptoms of cardiovascular diseases would increase the opportunity to use ARBs or ACE inhibitors. Normalization of the data with disease severity would be necessary. In addition, treatment with ACE inhibitors or ARBs should be compared with other anti-hypertensive drugs."

Please acknowledge that this point is conceptually related to points II-3 and II-4, and they have thus been discussed together. We entirely agree that the worse symptoms of cardiovascular diseases may increase the opportunity to use ARBs or ACE inhibitors, thus RAAS inhibitors on diabetes patients might have protective impacts on COVID-19 severity. And we thus agree that beneficial or adverse effects of the drugs should have been examined more rigorously.

Accordingly, in order to investigate whether RAAS inhibitors might have had protective impacts among the patients with diabetes, COPD and CVD, we stratified the analyses on the potential association between RAAS inhibitors and COVID-19 severity by CVD, COPD, and diabetes. Please find below the results on the sample of hypertensive subjects.

Antihypertensive treatment with: Mild Severe Very severe/lethal

Variables 

Stratified by CVD 

Without CVD (n=155) (n=106) (n=69)

- ACE inhibitors, % 54.2 45.1 49.3

- ARBs, % 41.9 46.2 42.0

- ACE inhibitors or ARBs, % 91.0 80.2 88.4

With CVD (n=52) (n=101) (n=60)

- ACE inhibitors, % 44.2 43.6 36.7

- ARBs, % 40.4 40.6 38.3

- ACE inhibitors or ARBs, % 80.8 77.2 71.7

Stratified by diabetes 

Without diabetes (n=165) (n=151) (n=88)

- ACE inhibitors, % 55.2 45.7 44.3

- ARBs, % 38.8 39.7 34.1

- ACE inhibitors or ARBs, % 89.1 78.8 76.2

With diabetes (n=42) (n=56) (n=41)

- ACE inhibitors, % 38.1 33.9 41.5

- ARBs, % 52.4 53.6 53.7

- ACE inhibitors or ARBs, % 85.7 78.6 90.2

Stratified by COPD 

Without COPD (n=194) (n=179) (n=110)

- ACE inhibitors, % 53.1 44.7 44.6

- ARBs, % 40.7 44.7 42.7

- ACE inhibitors or ARBs, % 88.7 81.6 83.6

With COPD (n=13) (n=28) (n=19)

- ACE inhibitors, % 30.8 28.6 36.8

- ARBs, % 53.9 35.7 26.3

- ACE inhibitors or ARBs, % 84.6 60.7 63.2

Among patients with diabetes, COPD or CVD, the use of RAAS inhibitors was not significantly associated with COVID-19 severity. We accordingly added the following sentences in the Discussion section: " In our sample, in order to further investigate the potential beneficial effects of ARBs and ACE inhibitors, the impact of which might be larger in patients with diabetes, COPD, or cardiovascular diseases, we performed additional analyses, stratified by comorbidities. We found however no significant differences in the risk of severe/lethal COVID-19 among treated and untreated patients with either CVD, or diabetes and COPD (data not shown)".

We also agree that it may be of interest to compare ACE inhibitors or ARBs with other anti-hypertensive drugs. However, unfortunately, we could not have the data, as the original protocol focused only on ARBs or ACE inhibitors, and the data on other anti-hypertensive drugs were therefore not collected.

II-3. The Referee wrote: “The authors compared with or without ARBs / ACE inhibitors in hypertension. Sometimes diseases are overlapped. They should clarify whether diabetes or COPD are overlapped with cardiovascular diseases. RAS inhibitors on diabetes patients might have protective impacts on COVID-19 severity."

We agree that some overlapping between cardiovascular diseases and diabetes or COPD exists. Also, we agree that the potential protective effect of RAAS inhibitors among the patients with diabetes or other comorbidities should have been investigated further (as suggested in the following point). We accordingly performed additional, stratified analyses in order to investigate whether RAAS inhibitors might have had protective impacts among the patients with diabetes, COPD and CVD, separately. Please acknowledge that this is issue is conceptually related to issues II-2 and II-3 and was discussed after point II-2.

Concerning the potential overlapping between CVD and COPD or diabetes, this was certainly the case, as shown below. However, please acknowledge that the overlapping was not as high as to cause multicollinearity. It was therefore possible to perform multivariable analyses, adjusting for the potential influence of CVD on the association between RAAS inhibitors and COVID-19 among the subjects with diabetes and COPD.

 | Diabetes

 CVD | 0 1 | Total

-----------+----------------------+----------

 0 | 1,224 121 | 1,345 

 1 | 185 73 | 258 

-----------+----------------------+----------

 Total | 1,409 194 | 1,603

 | COPD

 CVD | 0 1 | Total

-----------+----------------------+----------

 0 | 1,297 48 | 1,345 

 1 | 209 49 | 258 

-----------+----------------------+----------

 Total | 1,506 97 | 1,603

II-4. The Referee wrote: “It is interesting to see the ACE Inhibitors or ARBs treatment showing a trend for reduced odds ratio (Table 2). Beneficial or adverse effects of the drugs would have to be examined more rigorously. For instance, was prescription or medication continued during the infection for each patient?"

We entirely agree that we should have clarified this issue, and try to investigate further the potential beneficial effects of the drugs on COVID-19. With regard to medication, we added in the text that the treatment with ARBs or ACE inhibitors was not discontinued for any participant. Concerning the potential beneficial effect of the drugs, we performed additional, stratified analyses in order to investigate whether RAAS inhibitors might have had protective impacts among the patients with diabetes, COPD and CVD, separately. Please acknowledge that this is issue is conceptually related to issues II-2 and II-3 and was discussed after point II-2.

II-5. The Referee wrote: “For references, they cited recent short review or commentary on COVID-19 but mostly ignored key original papers. This is not acceptable. There are nearly two decades of research on ACE2 and SARS-CoV accumulated, and both basic and clinical papers of origin should be cited. The reference should be extensively revised."

We agree and thank the Referee for the suggestion. We accordingly replaced the references previously numbered as 5-6 with the followings (now numbered as [5-8]):

5. Hoffmann M, Kleine-Weber H, Schroeder S, et al. SARSCoV-2 cell entry depends on ACE2 and TMPRSS2 and is blocked by a clinically proven protease inhibitor. Cell 2020;181:271-80.

6. Kuba K, Imai Y, Rao S, Gao H, Guo F, Guan B, et al. A crucial role of angiotensin converting enzyme 2 (ACE2) in SARS coronavirus-induced lung injury. Nat Med. 2005;11(8):875-9.

7. Li W, Moore MJ, Vasilieva N, et al. Angiotensin-converting enzyme 2 is a functional receptor for the SARS coronavirus. Nature 2003;426: 450-4.

8. Xiao X, Chakraborti S, Dimitrov AS, Gramatikoff K, Dimitrov DS. The SARS-CoV S glycoprotein: expression and functional characterization. Biochem Biophys Res Commun. 2003;312(4):1159-64

Finally, please acknowledge that we also added the following references:

9. Furuhashi M, Moniwa N, Mita T, Fuseya T, Ishimura S, Ohno K, et al. Urinary angiotensin-converting enzyme 2 in hypertensive patients may be increased by olmesartan, an angiotensin II receptor blocker. Am J Hypertens. 2015;28(1):15-21.

10. Luque M, Martin P, Martell N, Fernandez C, Brosnihan KB, Ferrario CM. Effects of captopril related to increased levels of prostacyclin and angiotensin-(1-7) in essential hypertension. J Hypertens. 1996;14(6):799-805.

11. Zisman LS, Keller RS, Weaver B, Lin Q, Speth R, Bristow MR, et al. Increased angiotensin-(1-7)-forming activity in failing human heart ventricles: evidence for upregulation of the angiotensin-converting enzyme Homologue ACE2. Circulation. 2003;108(14):1707-12.

12. Ferrario CM, Jessup J, Chappell MC, Averill DB, Brosnihan KB, Tallant EA, et al. Effect of angiotensin-converting enzyme inhibition and angiotensin II receptor blockers on cardiac angiotensin-converting enzyme 2. Circulation. 2005;111(20):2605-10.

13. Gallagher PE, Ferrario CM, Tallant EA. MAP kinase/phosphatase pathway mediates the regulation of ACE2 by angiotensin peptides. Am J Physiol Cell Physiol. 2008;295(5):C1169-74.

14. Igase M, Strawn WB, Gallagher PE, Geary RL, Ferrario CM. Angiotensin II AT1 receptors regulate ACE2 and angiotensin-(1-7) expression in the aorta of spontaneously hypertensive rats. Am J Physiol Heart Circ Physiol. 2005;289(3):H1013-9.

15. Ishiyama Y, Gallagher PE, Averill DB, Tallant EA, Brosnihan KB, Ferrario CM. Upregulation of angiotensin-converting enzyme 2 after myocardial infarction by blockade of angiotensin II receptors. Hypertension. 2004;43(5):970-6.

16. Soler MJ, Ye M, Wysocki J, William J, Lloveras J, Batlle D. Localization of ACE2 in the renal vasculature: amplification by angiotensin II type 1 receptor blockade using telmisartan. Am J Physiol Renal Physiol. 2009;296(2):F398-405.

18. Esler M, Esler D. Can angiotensin receptor-blocking drugs perhaps be harmful in the COVID-19 pandemic? J Hypertens 2020;38:781-2.

19. Sommerstein R, Grani C. Preventing a COVID-19 pandemic: ACE inhibitors as a potential risk factor for fatal COVID-19. BMJ 2020;368: m810.

---

## [Decision Letter · Decision Letter 1]

4 Jun 2020

PONE-D-20-13065R1

Predictors of severe or lethal COVID-19, including Angiotensin Converting Enzyme Inhibitors and Angiotensin II Receptor Blockers, in a sample of infected Italian citizens.

PLOS ONE

Dear Dr. Manzoli,

Thank you for submitting your manuscript to PLOS ONE. After careful consideration, we feel that it has merit but does not fully meet PLOS ONE’s publication criteria as it currently stands. Therefore, we invite you to submit a revised version of the manuscript that addresses the points raised during the review process.

Please provide statistical power analysis as reviewer 2 requested.

We look forward to receiving your revised manuscript.

Kind regards,

Tatsuo Shimosawa, M.D., Ph.D.

Academic Editor

PLOS ONE

Reviewers' comments:

Reviewer's Responses to Questions

**Comments to the Author**

1. If the authors have adequately addressed your comments raised in a previous round of review and you feel that this manuscript is now acceptable for publication, you may indicate that here to bypass the “Comments to the Author” section, enter your conflict of interest statement in the “Confidential to Editor” section, and submit your "Accept" recommendation.

Reviewer #1: All comments have been addressed

Reviewer #2: (No Response)

2. Is the manuscript technically sound, and do the data support the conclusions?

Reviewer #1: Yes

Reviewer #2: No

3. Has the statistical analysis been performed appropriately and rigorously? 

Reviewer #1: Yes

Reviewer #2: N/A

4. Have the authors made all data underlying the findings in their manuscript fully available?

Reviewer #1: Yes

Reviewer #2: No

5. Is the manuscript presented in an intelligible fashion and written in standard English?

Reviewer #1: Yes

Reviewer #2: No

6. Review Comments to the Author

Reviewer #1: The authors have adequately responded to all queries I raised. The manuscript has been improved very much.

Reviewer #2: Comparison with other anti-hypertensive drugs is essential to precisely determine the effects of ACE inhibitors or ARBs on COVID-19 severity and lethality. In addition to COVID-19, the severity of underlying cardiovascular diseases should be also included. As such, it revealed that number and information of the patients in this study was not enough to reach their conclusion.

7. PLOS authors have the option to publish the peer review history of their article (what does this mean?). If published, this will include your full peer review and any attached files.

Reviewer #1: No

Reviewer #2: No

---

## [Author Response · Author response to Decision Letter 1]

6 Jun 2020

Answers to Editor's comments

E-1

The Editor wrote: "Please provide statistical power analysis as reviewer 2 requested". 

We agree and we accordingly added the following paragraph in the Methods section: "Our sample of 129 hypertensive subjects with severe/lethal COVID-19, and 414 hypertensive subjects with mild disease or asymptomatic infection, had 80% statistical power to detect a difference of 20% or higher (corresponding to a relative risk ≥1.20) in the risk of severe/lethal death between the users of ARBS or ACE inhibitors (exposed group), and non users (controls)".

Answers to Referee I comments

I-1. The Referee wrote: "The authors have adequately responded to all queries I raised. The manuscript has been improved very much".

We thank the Referee for his comment.

Answers to Referee II comments

II-1. The Referee wrote: "Comparison with other anti-hypertensive drugs is essential to precisely determine the effects of ACE inhibitors or ARBs on COVID-19 severity and lethality. In addition to COVID-19, the severity of underlying cardiovascular diseases should be also included. As such, it revealed that number and information of the patients in this study was not enough to reach their conclusion".

On one side, we agree that, hypothetically, both the severity of cardiovascular diseases and the use of other antihypertensive medications could be important to discern more precisely the relationship between ARBS or ACE inhibitors and COVID-19. On the other side, however, please acknowledge that:

1.

None of the four studies published on the topic measured the severity of underlying cardiovascular diseases (Mehta, JAMA Cardiol 2020; Mancia, N Engl J Med 2020; Reynolds, N Engl J Med 2020; Mehra, N Engl J Med 2020). Thus, this remains a plausible but unverified hypothesis and, as happened for the above studies, it cannot cancel the validity of the present study.

2.

Concerning the use of other antihypertensive drugs, of the four studies published on the topic, one did not report these data (Mehta, JAMA Cardiol 2020), one study reported the information but did not include this in multivariable analyses (Mehra, N Engl J Med 2020), and the other two (Mancia, N Engl J Med 2020; Reynolds, N Engl J Med 2020) collected these data and included them in multivariable analysis. In both cases, however, there were no substantial differences between the adjusted and unadjusted relative risks of death of ARBS or ACE inhibitors (which suggests that the additional covariates of the models, including other antihypertensive medications, did not impact alter noticeably the relationship between severity and use of ARBS or ACE inhibitors). This is also supported by the fact that our results were in line with those of the two studies from Mancia et al., and Reynolds et al. Therefore, also this hypothesis, although plausible, remains unverified.

Since it is not proven that both the severity of underlying cardiovascular diseases and use of other antihypertensive drugs may confound, or even mediate, the relationship between COVID-19 and use of ARBS or ACE inhibitors, the validity of the study cannot be nullified by the absence of an evaluation of these hypotheses. We do agree, however, that both hypotheses are potential limitations, and they should have been acknowledged in the text. Accordingly, the following sentences were added in the limitations section:

"Other limitations are the lack of an evaluation of the severity of the underlying cardiovascular diseases, and the absence of data on other antihypertensive medications. However, their potential role in altering the relationship between ARBS or ACE inhibitors and risk of severe/lethal COVID-19 remains unclear: none of the previous studies on the topic assessed cardiovascular diseases severity [26-29], and the two studies that included the use of other antihypertensive drugs into multivariable analyses did not find substantial differences between the adjusted and unadjusted relative risks of death [27,28]".

II-2. The Referee wrote: "The authors did not made all data underlying the findings in their manuscript fully available".

Please acknowledge that this is incorrect: we uploaded the entire dataset - named Appendix S1 - as online supporting information since the initial submission, to comply with PLOS One data availability requirement. Please acknowledge that, at the end of the manuscript, under the "Access to data" section, we reported the following sentence: "The complete dataset used in this work is available in Appendix S1".

---

## [Editor Report · Decision Letter 2]

10 Jun 2020

PONE-D-20-13065R2

Predictors of severe or lethal COVID-19, including Angiotensin Converting Enzyme Inhibitors and Angiotensin II Receptor Blockers, in a sample of infected Italian citizens.

PLOS ONE

Dear Dr. Manzoli,

Thank you for submitting your manuscript to PLOS ONE. After careful consideration, we feel that it has merit but does not fully meet PLOS ONE’s publication criteria as it currently stands. Therefore, we invite you to submit a revised version of the manuscript that addresses the points raised during the review process.

The authors responded adequately, however, the reference 26 is retracted.  Please edit the manuscript without citing this article.  Also the information of reference 27 is not completed.

We look forward to receiving your revised manuscript.

Kind regards,

Tatsuo Shimosawa, M.D., Ph.D.

Academic Editor

PLOS ONE

---

## [Author Response · Author response to Decision Letter 2]

10 Jun 2020

Answers to Editor's comments

E-1. The Editor wrote: "The authors responded adequately, however, the reference 26 is retracted. Please edit the manuscript without citing this article".

We agree and we thank the Editor for the comment. Accordingly, we deleted all the citations to the retracted article throughout the text. Please acknowledge that, in place of the above reference, we added the following new reference for some of the sentences in the Discussion:

44. Williamson E, Walker AJ, Bhaskaran K, Bacon S, Bates C, Morton CE, et al. OpenSAFELY: factors associated with COVID-19-related hospital death in the linked electronic health records of 17 million adult NHS patients. MedRxiv. 2020. doi: https://doi.org/10.1101/2020.05.06.20092999.

E-2. The Editor also wrote: "Also the information of reference 27 is not completed".

We agree and accordingly updated and completed the reference. Please accept our apologies for the oversight.

---

## [Editor Report · Decision Letter 3]

12 Jun 2020

Predictors of severe or lethal COVID-19, including Angiotensin Converting Enzyme Inhibitors and Angiotensin II Receptor Blockers, in a sample of infected Italian citizens.

PONE-D-20-13065R3

Dear Dr. Manzoli,

We’re pleased to inform you that your manuscript has been judged scientifically suitable for publication and will be formally accepted for publication once it meets all outstanding technical requirements.

Kind regards,

Tatsuo Shimosawa, M.D., Ph.D.

Academic Editor

PLOS ONE
---

## [Editor Report · Acceptance letter]

16 Jun 2020

PONE-D-20-13065R3 

Predictors of severe or lethal COVID-19, including Angiotensin Converting Enzyme Inhibitors and Angiotensin II Receptor Blockers, in a sample of infected Italian citizens. 

Dear Dr. Manzoli:

I'm pleased to inform you that your manuscript has been deemed suitable for publication in PLOS ONE. Congratulations! Your manuscript is now with our production department. 

Kind regards, 

on behalf of

Prof. Tatsuo Shimosawa 

Academic Editor

PLOS ONE